# Construction of a High-Density Genetic Map and Mapping of Firmness in Grapes (*Vitis vinifera* L.) Based on Whole-Genome Resequencing

**DOI:** 10.3390/ijms21030797

**Published:** 2020-01-25

**Authors:** Jianfu Jiang, Xiucai Fan, Ying Zhang, Xiaoping Tang, Xiaomei Li, Chonghuai Liu, Zhenwen Zhang

**Affiliations:** 1College of Enology, Northwest A&F University, Yangling 712100, China; jiangjianfu@caas.cn; 2Zhengzhou Fruit Research Institute, Chinese Academy of Agricultural Sciences, Zhengzhou 450009, China; fanxiucai@caas.cn (X.F.); zhangying05@caas.cn (Y.Z.); 3Pomology Institute, Shanxi Academy of Agricultural Sciences, Taigu 030815, China; txp-19590401@163.com (X.T.); gsslxm@163.com (X.L.)

**Keywords:** grape, high-density genetic map, quantitative trait loci, berry firmness, whole-genome resequencing

## Abstract

Berry firmness is one of the most important quality traits in table grapes. The underlying molecular and genetic mechanisms for berry firmness remain unclear. We constructed a high-density genetic map based on whole-genome resequencing to identify loci associated with berry firmness. The genetic map had 19 linkage groups, including 1662 bin markers (26,039 SNPs), covering 1463.38 cM, and the average inter-marker distance was 0.88 cM. An analysis of berry firmness in the F1 population and both parents for three consecutive years revealed continuous variability in F1, with a distribution close to the normal distribution. Based on the genetic map and phenotypic data, three potentially significant quantitative trait loci (QTLs) related to berry firmness were identified by composite interval mapping. The contribution rate of each QTL ranged from 21.5% to 28.6%. We identified four candidate genes associated with grape firmness, which are related to endoglucanase, abscisic acid (ABA), and transcription factors. A qRT-PCR analysis revealed that the expression of abscisic-aldehyde oxidase-like gene (*VIT_18s0041g02410*) and endoglucanase 3 gene (*VIT_18s0089g00210*) in Muscat Hamburg was higher than in Crimson Seedless at the veraison stage, which was consistent with that of parent berry firmness. These results confirmed that *VIT_18s0041g02410* and *VIT_18s0089g00210* are candidate genes associated with berry firmness.

## 1. Introduction

Grape (*Vitis vinifera* L.) is one of the most economically important fruit-tree crops in the world, with a global production of 74 million tons in 2017 (available online: http://www.fao.org/faostat). Grape berries are widely used as table grapes and to produce wine, raisins, and juice [1]. Grapes have a high nutritional value and provide health benefits [2]. Berry firmness is one of the main factors affecting consumers’ acceptance [3,4,5]. Therefore, it is an important trait in table grape breeding [6,7]. High berry firmness is associated with good shelf-life performance and less postharvest losses, which is especially important for year-round fruit marketability and shipping overseas [8]. Therefore, breeding and propagating new cultivars with high firmness are very important for grape production.

Grapevines are woody perennials with a high degree of heterozygosity, and it generally takes 4 to 5 years for a seed to grow and develop into a fruit [9,10]. Traditional grape breeding to obtain a plant with all the desired features is time-consuming, laborious, and expensive [11,12,13,14]. With the development of modern molecular biology, plant breeders now use marker-assisted selection (MAS) for grape breeding. By screening molecular markers related to target traits, hybrid progenies can be selected at the early seedling stage; therefore, MAS can greatly shorten the breeding process and improve breeding efficiency [15,16].

Many important economic traits in grapes, such as yield, quality, and resistance, are quantitative traits; and their phenotypes are continuously distributed in the progeny [17,18,19]. There is not a one-to-one correspondence between the phenotype and genotype. A quantitative trait locus (QTL) analysis based on linkage maps and phenotypic evaluation of segregating progenies has been widely used to investigate the genetic determinants of agronomic traits [20].

A (high-density) genetic map is essential for QTL mapping [21]. In the past ten years, a number of grape genetic maps have been constructed based on different mapping populations [20,22,23,24,25,26,27,28,29,30,31,32,33]. Numerous QTLs for important economic traits, including resistance to downy mildew [10,16,22,34,35,36,37,38], powdery mildew [34,36,39,40,41], anthracnose [24], root-knot nematodes [42], and grape phylloxera [43,44], as well as flower sex [45,46], berry color [46,47], seedlessness [48], berry weight [18,32], soluble solid content [32], acidity [20], and muscat flavor [49], have been identified.

Grape firmness, like most characteristics of agricultural interest, is a complex quantitative trait. Multiple QTLs for berry firmness have been identified in recent years. The first study for QTLs associated with berry firmness in table grapes found seven genomic regions on linkage groups (LGs) 1, 4, 5, 9, 10 (bottom region), 13, and 18, which individually explained up to 19.8% of the total phenotypic variance [6]. A second QTL study identified firmness determinants distributed in LGs 8 and 18, which explained 27.6% of the phenotypic variance, with confidence intervals up to 10 cM [50]. A third study using 98 F1 individuals from a *Vitis labruscana* × *V. vinifera* cross reported two QTLs for firmness that were located on LGs 3 and 10, with the former being more stable [18]. The identification of numerous polymorphic markers is an important prerequisite for constructing a genetic map [51,52]. The total number of markers in the LGs of maps generated in these previous studies is less than 500, and thus, the resolution of these genetic maps is low, which has limited the efficiency and accuracy of QTL mapping. Studies have demonstrated that the resolution of a genetic map can be significantly improved by increasing marker density [53,54]. Thus, it is necessary to construct a genetic linkage map using high-density molecular markers to improve the accuracy of QTL mapping for firmness-related traits in grapes.

Single-nucleotide polymorphisms (SNPs), as the third generation of molecular markers, have great advantages over other molecular markers in genomic and genetic studies. As SNPs are the most abundant heritable variations in genomes and can be detected and genotyped automatically and in a high-throughput manner, they have revolutionized high-quality genetic map construction [55,56]. Next-generation sequencing is a high-throughput, low-cost technology that has been used to identify a large number of SNPs for high-density genetic map construction in various crops, such as soybean [57], cotton [58], and pear [59]. In recent years, several genetic maps for grapevine have been constructed on the basis of high-throughput sequencing technology; these maps have increased marker density and have led to the identification of some new QTLs [10,24,28,33,46]. Nearly all of the genetic maps were constructed using reduced representation methods, such as restriction site-associated DNA sequencing and genotyping-by-sequencing, which are relatively cheap as they sample only a fraction of the genome, but they also produce incomplete data [60]. With the development of high-throughput sequencing technologies and the availability of a reference genome for grapevines [61], whole-genome resequencing (WGR) allows the identification of whole genome differences between individuals and large numbers of SNPs, and has become one of the most rapid and effective methods used in QTL mapping and breeding research [62,63]. 

This study aimed to identify candidate regions and genes for berry firmness in grapevines to facilitate MAS. To this end, we used an F1 population consisting of 105 individuals to construct a high-resolution genetic map based on WGR. The F1 population was derived from a cross between Muscat Hamburg and Crimson Seedless (MH × CS), which have significantly different berry firmness. 

## 2. Results

### 2.1. Phenotypic Analysis of Berry Firmness

The values of berry firmness measured in the parents (female: MH, male: CS) and F1 population over three years (2016–2018) are shown in Table 1. Berry firmness was significantly different between the parents in all three years, with the firmness of CS being significantly higher than that of MH.

An analysis of berry firmness in the offspring indicated that there were a certain number of “super-parent” individuals in the population (Figure 1). Berry firmness in the offspring showed continuous variation, with a distribution that was close to the normal distribution, indicating that this trait is quantitatively inherited and controlled by multiple genes.

### 2.2. WGR of the F1 Population and the Two Parents

The HiSeq 4000 sequencing platform was used for double-terminal WGR of the parents and F1 population. After filtering the raw reads and removing low-quality sequences and redundant and unpaired reads, 5.27 Gb of clean reads and 790.63 Gb of clean data were obtained. The Q30 reads ranged between 91.96% and 94.05%, with an average of 93.20%, indicating good sequencing data quality. The GC content was between 36.20% and 38.37%, with an average of 36.82%, and was normally distributed (Appendix A).

The clean reads were aligned to the grape reference genome (PN40024 assembly 12X) using the Burrows–Wheeler Aligner software [61]. In total, 96.30% of the total effective bases were mapped for the female parent, 95.48% for the male parent, and 96.32% for the offspring. The fraction of bases mapped to unique genome positions was 71.76% for the female parent, 70.39% for the male parent, and 70.86% for the offspring. The genome coverage and effective sequencing depth were 94.26%, and 44.93×; 93.49% and 47.05×; and 90.86–92.95%, and 11.67–20.22× for the female parent, male parent, and progeny, respectively (Appendix A). Thus, the sequencing data evenly covered the genome.

### 2.3. SNP Marker Identification

Based on the genotype data of the two parents, SNP markers were identified. The sites, for which parental information was missing, were filtered out. Thus, we identified 27,695 SNPs, which were classified into three segregation patterns (Table 2); the main patterns were lm × ll and nn × np, and a small number of hk × hk markers was found.

A chi-square test of the 27,695 SNP markers in the F1 population revealed that the genotype frequency of 1656 (5.98%) loci significantly deviated from the expected Mendel frequency (*p* < 0.1; Table 3). Among the partially segregated markers, 1620 markers did not conform to 1:1 segregation (lm × ll and nn × np), 36 markers did not conform to 1:2:1 segregation, and most were biased toward the genotype of female or male parent. The 1656 partially segregated markers were eliminated, and the remaining 26,039 SNP markers were used to construct genetic maps.

### 2.4. Construction of a High-Density Genetic Map

A genetic map was constructed using the Lep-MAP3 software (https://sourcefrorge.net/projects/lep-map3/) and 26,039 SNP markers. The software uses the maximum-likelihood method. The filtered markers were clustered using various log of odds ratio (LOD) values. It was found that when LOD = 7, the clustering result was ideal, with 19 major LGs and good correspondence with chromosome information. The Kosambi algorithm was used to sort the markers in each group and calculate the genetic distances.

The genetic map of the female parent (MH) contained 935 bin markers (12,949 SNPs), spanning 1494.56 cM, with an average inter-marker distance of 1.60 cM (Appendix A). LG16 (86) and LG6 (14) had the most and least markers, respectively; the largest gap was on LG3 (46.52 cM) and the smallest gap was on LG19 (2.89 cM). The “Gap < 5 cM” percentage for each LG ranged from 95.24% (LG8) to 100% (LG6, LG7, LG15, LG16, and LG19). LG5 was the longest (102.46 cM), whereas LG6 was the shortest (16.36 cM). The largest average inter-marker distance was on LG3 (3.87 cM) and the smallest average distance was on LG16 (1.15 cM).

The genetic map of the male parent (CS) contained 913 bin markers (13,257 SNPs), spanning 1494.67 cM, with an average inter-marker distance of 1.64 cM (Appendix A). LG16 (76) and LG8 (30) had the most and least markers, respectively. The largest gap was on LG3 (35.88 cM) and the smallest gap was on LG2 (3.85 cM). The “Gap < 5 cM” percentage ranged from 94.81% (LG8) to 100% (LG12 and LG2). LG16 was the longest (105.90 cM), whereas LG11 was the shortest (56.83 cM). The largest average inter-marker distance was on LG5 (2.51 cM) and the smallest distance was on LG12 (1.23 cM).

The integrated map contained 1662 bin markers (26,039 SNPs), spanning 1460.38 cM (Figure 2), with 87.47 bin markers per LG on average and an average inter-marker distance of 0.88 cM (Table 4). The length of LGs ranged from 14.91 cM (LG17) to 100.53 cM (LG18), with an average length of 76.86 cM. LG16 contained the most bin markers (152) with an average genetic interval of 0.59 cM, whereas LG17 contained the least bin markers (20). The average “Gap < 5 cM” percentage was 99.89%. The largest average inter-marker distance was on LG1 (1.18 cM) and the smallest distance was on LG16 (0.59 cM).

### 2.5. QTLs for Berry Firmness

We identified three QTLs (LOD ≥ 4.0) related to berry firmness from the integrated genetic map for the MH × CS F1 population and berry firmness data for the population for three consecutive years by using the MapQTL6.0 software with the interval mapping method. The three QTLs were located on LG 18 and named *qBF18-2016*, *qBF18-2017*, and *qBF18-2018*; and the maximum LOD scores were 4.98, 4.88, and 5.86, which explained 22.3%, 21.5%, and 28.6% of the phenotypic variation, respectively (Table 5). The genetic positions were 67.81–68.29 cM, 59.64–60.60 cM, and 57.23–61.56 cM, and corresponded to the physical positions 26.58–27.20 Mb, 25.03–28.59 Mb, and 24.64–28.59 Mb of chromosome 18, respectively (PN40024 assembly 12X) [61] (Figure 3).

### 2.6. Prediction of Candidate Genes Related to Berry Firmness

Based on the markers in the segment and their physical positions on the grape genome (PN40024 assembly 12X), the candidate gene segments were predicted. There were 22, 217, and 244 (total 244) genes in each QTL region corresponding to the grape genome (position chr18: 24639353–28587457) (Appendix A). Previous studies have shown that cell wall enzymes and plant hormones play an important role in berry softening [64,65,66,67]. Gene function annotations of this segment were used to exclude unrelated genes and an analysis using Gene Ontology (GO), Kyoto Encyclopedia of Genes and Genomes (KEGG), and NCBI non-redundant protein database (Nr) was performed. The analysis revealed four candidate genes related to berry firmness (Table 6). Specifically, we identified one endoglucanase gene (*VIT_18s0089g00210*), one gene related to abscisic acid (ABA) (*VIT_18s0041g02410*), and two transcription factor genes (*VIT_18s0041g00700* and *VIT_18s0041g02140*) (Table 6).

Furthermore, an expression analysis of the four candidate genes was performed using qRT-PCR for berry firmness at three developmental stages. As shown in Figure 4, berry firmness of MH decreased significantly at veraison, while that of CS decreased significantly at the maturity stage. The expression levels of *VIT_18s0089g00210* and *VIT_18s0041g02410* were higher in MH than in CS at the veraison stage, and the opposite trend was observed in the maturity stage. Notably, there were no obvious changes in the expression of the other studied genes. The expression patterns of *VIT_18s0089g00210* and *VIT_18s0041g02410* were consistent with that of parent berry firmness, suggesting that *VIT_18s0089g00210* and *VIT_18s0041g02410* are likely the candidate genes associated with berry firmness of grapes.

## 3. Discussion

Berry firmness is one of the commercial qualities of grapes and an important standard in table grape breeding [68]. The reliability of the berry firmness identification method is of great significance in grape QTL mapping research. For assessing the textural features of a grape, both sensory and instrumental analyses can be used [18,69,70]. Traditionally, sensory tasting by trained judges is often subjective and more complicated when the number of samples is large or if the differences between samples are small [71]. Texture profile analysis (TPA) is a well-developed and reliable method used for evaluating the textural characteristics of foods and fruits. The principle of this technique is to simulate the chewing movement of the human oral cavity, perform the compression process of the test sample twice, and output the texture parameters through the software. These parameters have been shown to be well-correlated with sensory evaluation of textural parameters [72]. Giacosa et al. used a texture analyzer to measure the flesh firmness of the five reference table grape cultivars mentioned in the ampelographic descriptor for grapes (OIV code 235), and considered that soft, lightly soft, and firm levels were defined by 0.074–0.117, 0.121–0.158, and 0.205–0.391 N/mm, respectively [73]. Conner et al. evaluated the berry firmness of 26 muscadine grape cultivars and showed that berry firmness of muscadine grapes ranged from very soft to firm, but still remained softer than *V. vinifera*. It is recommended to use a texture analyzer rather than sensory evaluation of grape berry firmness, because instrument evaluation is controllable under laboratory conditions, is easier, and provides more accurate results [74]. In this study, berry firmness of parents and offspring was evaluated by texture profile analysis using a TA-XT2i texture analyzer. The firmness of CS was significantly higher than that of MH. Moreover, there were a certain number of “super-parent” individuals in the population, and berry firmness in the offspring showed continuous variation. The distribution was close to the normal distribution, indicating that this trait is quantitatively inherited and controlled by multiple genes, which is consistent with previous studies [6,50].

A reliable genetic map is essential for identifying QTLs of traits of interest and prediction of candidate genes [75]; however, it is difficult to generate a hybrid population that is suitable for QTL mapping, such as recombinant inbred lines or F2 population in grapes and other fruit trees, which are all highly heterozygous, have long breeding cycles, and often have a smaller population size than those of annual crops [59,76]. In general, increasing the density of markers can increase the resolution of genetic maps, thereby improving the precision of QTL mapping [54,77]. Previous QTL studies of grape traits were mainly based on lower marker numbers (<1000) with relatively high QTL intervals, which affected the accuracy of QTLs and hard-to-locate candidate genes [33,78]. With the rapid development of sequencing technologies and bioinformatics, a large number of polymorphic molecular markers to meet the needs for high-density genetic map construction can be identified, and high-resolution linkage maps have been successfully used for QTL fine mapping in many crops, such as soybean, cotton, pear, and jujube [57,58,59,76]. In a previous grape study, Wang et al. constructed the first high-density genetic map spanning a genetic distance of 1917.3 cM and with an average distance of 1.16 cM between markers [31]. Sapkota et al. detected a major QTL for downy mildew resistance based on a high-resolution linkage map with 3825 markers in the cross-hybridization between Norton and Cabernet Sauvignon, which had 159 progenies, spanning a genetic distance of 2203.5 cM, and an average distance of 1.1 cM between markers [10]. Based on the F1 population (91 offspring), Fu et al. constructed a high-density genetic map, which covered a total of 1665.31 cM in length, with an average of 1.81 cM between markers. The authors detected a major stable QTL for ripe rot resistance (*Cgr1*) by using the constructed genetic map [24]. In this study, a high-density linkage genetic map was constructed via the whole-genome resequencing method, and a total of 790.63 Gb clean data were generated. The effective sequencing depths of the female parent, the male parent, and their progeny were 44.93×, 47.05×, and 11.67–20.22×, respectively, which were sufficient to detect an appreciable number of recombination breakpoints. The genetic map contained 1662 bin markers (26,039 SNPs), with a total genetic length of 1460.38 cM and an average length of 0.88 cM for adjacent markers. The length of LGs ranged from 14.91 cM to 100.53 cM, with an average length of 76.86 cM. The average “Gap < 5 cM” percentage was 99.89%. The average marker distance ranged from 0.59 cM to 1.18 cM, and the map density was higher than other published maps [20,24,26,29]. The linkage map developed in this study possessed good quality and can therefore be used for QTL mapping.

Compared to research on traits such as disease resistance, stress resistance, sugar content, and color, fruit firmness has been rarely studied, and mostly in cherry, apple, peach, tomato, and melon. For example, a major fruit firmness QTL termed qP-FF4.1 was identified in three sweet cherry populations, the candidate genes were associated with cell wall modification and various hormone signaling pathways, and the extended protein gene, especially, was the most promising candidate gene [64]. In apple, a QTL site termed Md-PG1, associated with apple hardness, which is regulated by ethylene, was identified based on an F1 mapping population generated from Fuji and Mondial Gala [79]. Ogundiwin et al. used a hybrid of different varieties of peach varieties to construct a linkage map, and QTLs related to texture quality were located on LGs 1, 4, 5, 7, and 8, and encoded pectinase, pectin methylesterase, and endopolygalacturonase. In tomato, a firmness QTL with five distinct subpeaks was identified, and genes coding for ethylene response factor and pectin methylesterase were nominated as QTL candidate genes [80]. Several QTLs for fruit flesh firmness have been identified in melon, and some candidate genes were speculated to be related to ethylene regulation, biosynthesis, and perception, and cell wall degradation [81]. In grapes, ninety-eight individual progenies of a cross between 626-84 and Iku82 were used to identify QTLs for grape berry traits, including cracking, weight, firmness, harvest time, seed number, difficulty of breakdown, and soluble solids. Berry firmness was categorized as soft, medium, slightly firm, or very firm. The results of this study showed that the difficulty of breakdown and berry firmness were positively correlated (r = 0.542). Two loci related to berry firmness were located in LG3 and LG10. The LG3 locus was linked to the primer VMC2E7, the LG10 locus was close to the middle region of the primers VVIH01 and UDV073, and the LG3 locus was more stable. A genotyping-by-sequencing approach to analyze the fruit traits of 179 grape varieties, including berry color, firmness, and flavor, revealed two sites related to berry firmness, located on chromosomes 16 and 17 and encoding mainly proteins associated with calcium channels [82]. Based on MH × Sugraone and Ruby Seedless × Moscatuel mapping populations, seven QTLs related to grape berry firmness, located on LGs 1, 4, 5, and 9 on chromosomes 10, 13, and 18, have been identified [6]. Later, a Ruby Seedless × Sultanina mapping population was used to identify two QTLs for berry firmness on chromosomes 8 and 18. The QTL on chromosome 8 was located between the markers UDV125 and VMCNG2H2 and explained approximately 15.6% of the phenotypic variation, and the QTL on chromosome 18 was located between VVIN16 and VVCS1E103N17FM1 and explained approximately 12% of the phenotypic variation. This was the first report of a QTL for grape berry firmness that was stable in different seasons [50]. In this study, we identified three QTLs related to berry firmness located on LG 18, which explained 21.5%–28.6% of the phenotypic variation. The genetic positions were 67.81–68.29 cM, 59.64–60.60 cM, and 57.23–61.56 cM, and corresponded to the physical positions 26.58–27.20 Mb, 25.03–28.59 Mb, and 24.64–28.59 Mb of chromosome 18, respectively. These QTLs are similar to those found by Carreño et al. [6] and Correa et al. [50] on LG18. The physical position of the SNPs did not coincide with genetic position in this study, which was likely attributed to errors in the alignment of reference genome sequence, as well as different microstructures on chromosomes [10].

Softening was one of the main characteristics for fruit ripening and senescence. During fruit ripening and softening, cell wall components such as pectin, cellulose, and hemicellulose are degraded, and the microfibril filament structure of the cell wall is loosened and softened. Moreover, ripening leads to the disappearance of intercellular filaments, thinning of the cell wall, and cell dispersal, which results in destruction of the cell wall structure and fruit softening. Previous studies have shown that cell wall enzymes and plant hormones play an important role in this process [64,65,66,67]. Therefore, genes related to the metabolism of plant cell wall or various hormone signaling pathways were all considered as candidate genes for this study. Four candidate genes were obtained through functional annotation prediction. Glucanase is one of the key cell wall hydrolytic enzymes, which can promote the random hydrolysis of β-1,4-glycosidic bonds to cleave cellulose into smaller fragments and loosen the cell wall [83]. Moreover, increased gene expression of glucanase has been associated with fruit ripening in bananas, pears, strawberries, and grapes [84,85,86,87,88]. The plant hormone, abscisic acid (ABA), plays a crucial role in fruit ripening and responses to environmental stresses [89]. The application of ABA during the grape growing period will soften the grape berry texture of Red Globe, Flame Seedless, and Crimson Seedless [90,91,92], suggesting that ABA is a major regulator of grape berry ripening onset [93,94]. It is likely that the expression of ABA genes was associated with changes in berry firmness. Transcription factors, such as MADS-box and NAC, have also been reported to be involved in fruit ripening of bananas, tomatoes apples, grapes, etc. [95,96,97,98]. Previous studies highlight that berry softening occurs primarily during the veraison stage, the stage in which berry firmness decreases sharply in the soft flesh variety [99]. *VIT_18s0089g00210* and *VIT_18s0041g02410*, annotated to endoglucanase 3 and abscisic-aldehyde oxidase-like genes, were involved in cell wall assembly during cell elongation and ABA biosynthesis, respectively [100,101]. The expression levels of these two genes were consistent with that of parent berry firmness. Thus, *VIT_18s0089g00210* and *VIT_18s0041g02410* are likely the candidate genes associated with berry firmness of grapes. Notably, the genes were located in stable candidate regions with high LOD values in this study. Given that there was no direct evidence that these genes control these characteristics, further experiments will be required to verify the function of these candidate genes. An increase in population size can increase the number of recombination events of offspring, which is conducive to improve QTL precision [102,103], The F1 population in this study was relatively small (105 offspring) and the population size can be increased to delimit QTL interval in future research. Our results provide valuable tools for future candidate gene identification, map-based gene cloning, and MAS for grape berry firmness.

## 4. Materials and Methods 

### 4.1. Plant Material and DNA Extraction

An F1 population of 105 individuals derived from a cross between Muscat Hamburg (*V. vinifera* L.) and Crimson Seedless (*V. vinifera* L.) was generated in May 2004. Muscat Hamburg, with soft berries, was used as the female parent, and Crimson seedless, with firm berries, was used as the male parent. Hybrid seeds were sown in a vineyard in March 2005. The seedlings began to bear fruit in the autumn of 2008. The population originally consisted of 133 individual plants. After false hybrids were removed using an simple sequence repeat (SSR) marker method, 105 true hybrid individuals with fruits were finally selected as the mapping population.

The seeds of the mapping population were sown in the vineyard of the Pomology Research Institute, Shanxi Academy of Agricultural Sciences (Shanxi Province, P.R. China; 37°23’N, 112°32’E). The vineyard soil is sandy loam and silty loam. The vineyard soil pH is 7.8. The average annual temperature in this region is 10.6 °C, annual sunshine duration is 2300 h, annual rainfall is 400–600 mm, frost-free period is 160–180 days, and effective accumulative temperature is 3675 °C [104]. The vines were planted in north–south orientation, with 0.8 m between individuals in a row and 2.5 m between rows. All vines were trained to a slope trunk with a vertical shoot positioning trellis system [105,106]. Clusters were thinned to 120–150 berries per bunch. Water and fertilizer were controlled using conventional management practices.

Young healthy leaves were harvested from the two parent plants and each individual progeny plant (F1 generation). The samples were immediately frozen in liquid nitrogen and stored at −80 °C until DNA extraction. Genomic DNA was extracted using an improved cetyltriethylammnonium bromide (CTAB) method [107]. The DNA was quantified with a NanoDrop 1000 spectrophotometer (NanoDrop, Wilmington, DE, USA) and evaluated by electrophoresis in a 0.8% agarose gel.

### 4.2. Phenotypic Evaluation

The maturity date was determined as reported by Correa et al. [50]. After veraison, the soluble solid content (SSC) of berries was monitored weekly, until it stabilized to 18 Brix or the seed color changed to dark brown. Three replicates, consisting of two clusters each, were randomly sampled at maturity, and five healthy and homogenous berries were sampled from the middle of each cluster. Berry firmness was evaluated by texture profile analysis using a TA-XT2i texture analyzer (Stable Micro Systems, Godalming, Surrey, UK). Each berry was peeled and individually compressed in the equatorial position using a 100-mm P/100 flat cylindrical probe. The operating parameters were as follows: pre-test speed = 2 mm/s, test speed = 1 mm/s, post-test speed = 1 mm/s, compression degree = 25%, time = 2 s, and trigger force = 5.0 g [73,108]. Firmness is defined as the peak force during the first compression of the sample [109]. Phenotypic evaluation was conducted for three consecutive years (2016–2018).

### 4.3. Phenotypic Data Analysis

Phenotypic data were statistically analyzed using SPSS version 13.0 (SPSS, Chicago, IL, USA). Significant differences between the mean values of parental traits were analyzed using a *t-*test. The mean, standard deviation (SD), kurtosis, and skewness values in the F1 population were calculated.

### 4.4. Sequencing Library Construction and High-Throughput Sequencing

Genomic DNA was sheared using a high-performance ultrasonicator (Covaris, Woburn, MA, USA). Short DNA fragments were obtained by adjusting the instrumental parameters. Sequencing libraries were constructed by terminal repair, followed by the addition of 3’A and a sequencing linker. The DNA was then purified and amplified by PCR. The HiSeq 4000 system (Illumina, San Diego, CA, USA) was used for paired-end sequencing [51].

### 4.5. SNP Identification and Genotyping

Raw reads were trimmed for adapter contamination and low-quality bases using SOAPnuke v.1.5.2 (https://github.com/BGI-flexlab/SOAPnuke) [110]. The clean reads were aligned to the grape reference genome (ftp.ensemblgenes.org/pub/plants/release38/fasta/vitis_vinifera/dna/vitis_ vinifera.iggp_12x.dna.toplevel.fa.gz.) using the Burrows–Wheeler Aligner program [61,111]. Genome Analysis Toolkit v.3.7 (Cambridge, MA, USA) was used to identify candidate SNPs among the lines [112]. SNPs of all individuals were integrated, and high-quality genotypic markers of the whole population were obtained through filtering using the following criteria: if the mass value of the genotype is less than 40 and the depth is less than 10 during SNP integration, the genotype is recorded as a deletion, marked as “-“, and marker sites with a genotype deletion rate of more than 20% of the sample are filtered out; markers at heterozygous sites or at the same homozygous sites in the parents are filtered out (i.e., only homozygous sites with differences in the two parents are retained); and genotype sites that are partially separated in the offspring are filtered out (screened by chi-square test with a significance level of *p* < 0.1). The filtered SNP markers were used for map construction.

### 4.6. Genetic Map Construction

The genetic map was constructed using the Lep-MAP3 software (https://sourcefrorge.net/projects/lep-map3/), which uses the maximum-likelihood method [113]. Attempts were made to cluster the filtered markers with LOD values of different gradients. It was found that when LOD = 7, the clustering result was ideal, with 19 major linkage groups and good correspondence with chromosome information of the genome. The Kosambi algorithm was used to sort the markers of each group and calculate genetic distances [114].

### 4.7. QTL Mapping and Candidate Gene Prediction

QTL mapping was performed on the population using the integrated map results and phenotypic data, and the multiple QTL model was implemented in the MapQTL 6.0 software [115,116]. The results were screened for functional QTLs based on an LOD value ≥ 4.0 [49]. The QTLs were named according to the recommended nomenclature system [117]. Annotations from the GO, KEGG, and Nr databases were used to categorize the candidate genes [51].

### 4.8. RNA Isolation and qRT-PCR Analysis of the Candidate Genes

The candidates for berry firmness were further analyzed by qRT-PCR. The peeled berry sample from three fruit developmental stages (green stage, veraison and maturity) of CS (30, 90, and 110 days after flowering) and HS (30, 70, and 90 days after flowering) were collected and each grapevine was used as a biological replicate [118]. Total RNA was isolated from peeled berry samples using the procedure outlined in a previous study [119]. The first-stand cDNA was synthesized using a PrimeScript™ RT reagent kit (TaKaRa, Dalian, China) according to the manufacturer’s instructions. *β-Actin* was selected as an internal reference gene [120]. The qRT-PCR validation was performed with a StepOne^TM^ real-time PCR system (Applied Biosystems, Thermo Fisher Scientific, Foster City, CA, USA). All the primers were designed with Premier 5.0 software and sequenced by Sangon Biotech (Shanghai) Co., Ltd. All reactions were performed in three independent biological replicates, and the relative gene expression was analyzed using the 2^−ΔΔCt^ method [121]. The primer sequences used in this study are listed in Appendix A.

## Figures and Tables

**Figure 1 ijms-21-00797-f001:**
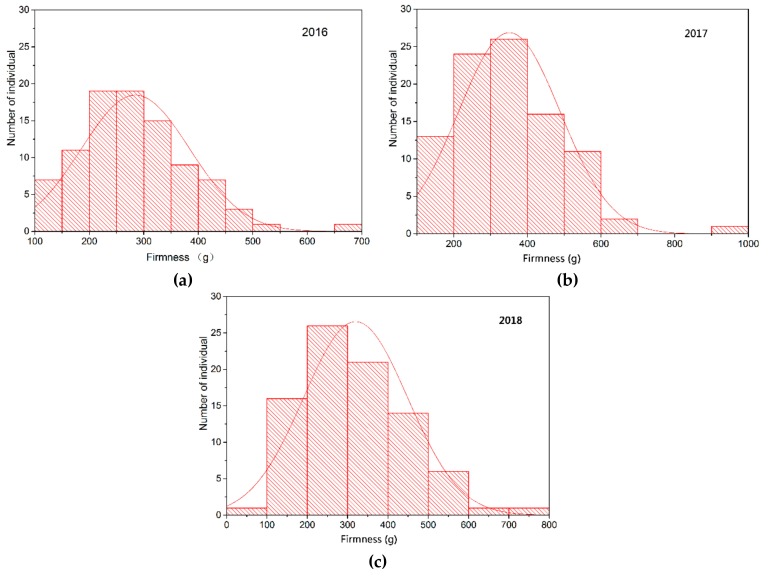
Frequency distribution of berry firmness in the F1 population in (**a**) 2016, (**b**) 2017, and (**c**) 2018.

**Figure 2 ijms-21-00797-f002:**
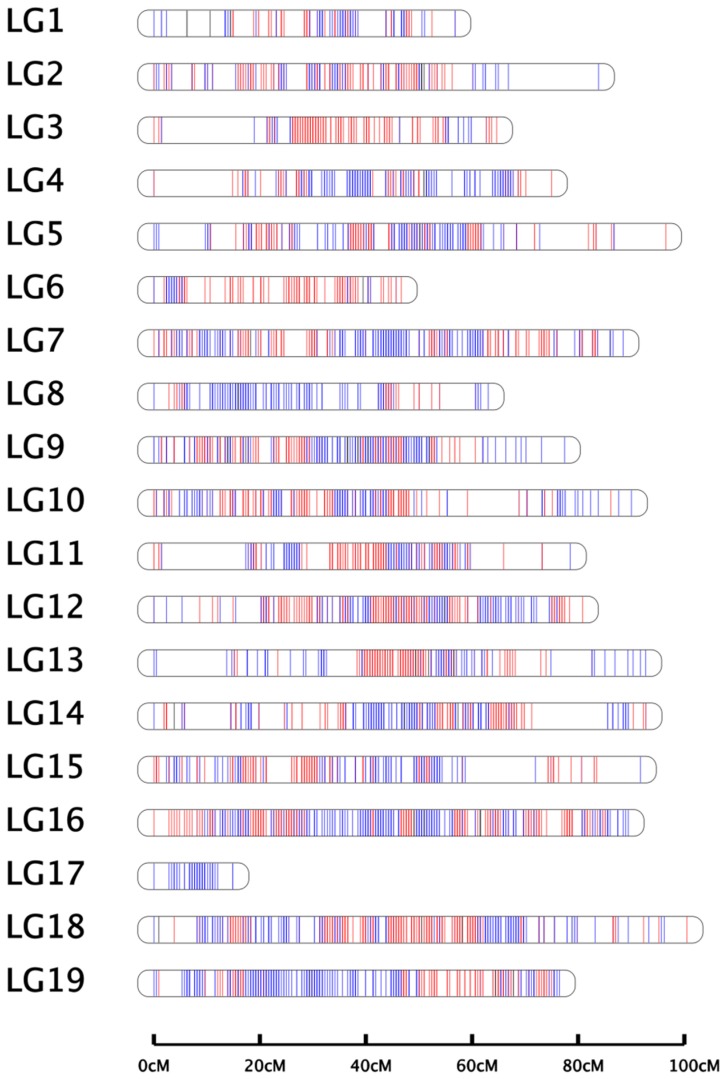
Integrated linkage groups for Muscat Hamburg × Crimson Seedless. Paternal sites are indicated in red, maternal sites are indicated in blue, heterozygous loci are indicated in black. The linkage groups were numbered according to acknowledged references (PN40024 assembly 12X) [61].

**Figure 3 ijms-21-00797-f003:**
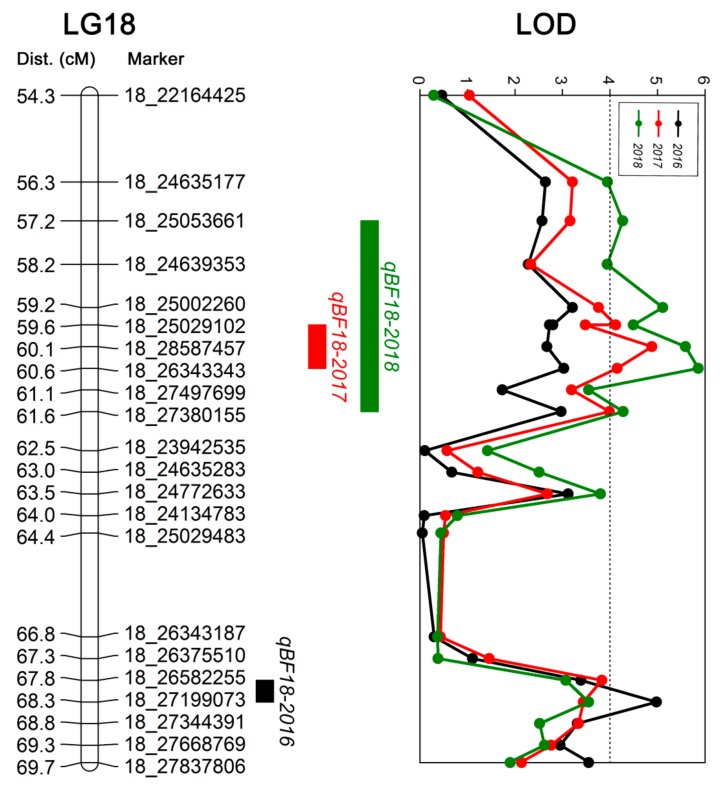
Quantitative trait loci (QTL) of berry firmness on the genetic linkage map of three years.

**Figure 4 ijms-21-00797-f004:**
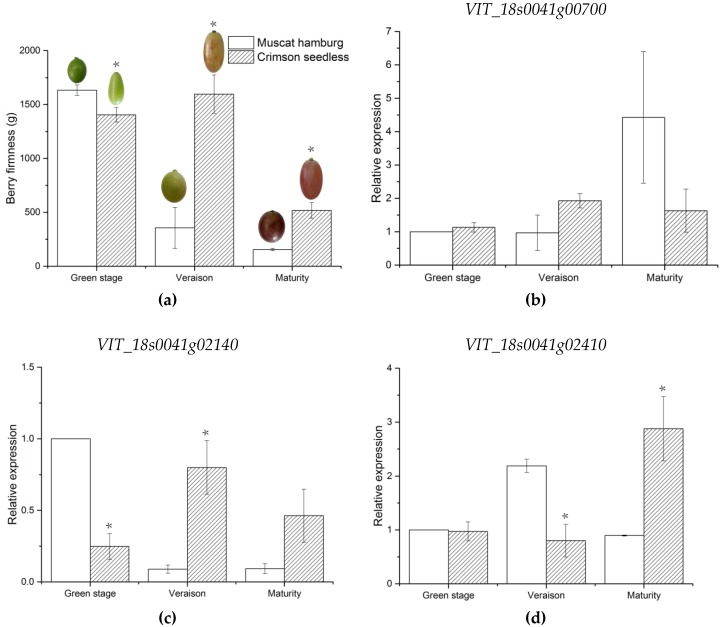
Changes in berry firmness (**a**) and qRT-PCR analysis of the expression of four candidate genes(**b**–**e**) in parents at three fruit developmental stages. Asterisk indicates significant differences by *t*-test at *p* < 0.05. Data are the mean ± SD of three replications.

**Table 1 ijms-21-00797-t001:** Descriptive statistics of berry firmness for the parents and the F1 population.

Year	Parents		F1 Population				
	MH	CS	Min.	Max.	Mean	Skewness	Kurtosis
2016	192.04 ± 11.36 ^a^	621.20 ± 63.20 ^b^	118.87	665.28	284.62	0.75	1.23
2017	212.71 ± 41.35 ^a^	634.71 ± 96.60 ^b^	142.21	956.65	350.92	1.14	2.80
2018	154.16 ± 42.11 ^a^	650.37 ± 72.78 ^b^	86.49	727.19	319.79	0.70	0.51

^a,b^ Different letters indicate a significant difference between the two parents.

**Table 2 ijms-21-00797-t002:** Number of markers in each of the segregation patterns.

Segregation Pattern	Segregation Ratio	Marker Number
lm × ll	1:1	13,876
nn × np	1:1	13,607
hk × hk	1:2:1	212
Total markers		27,695

**Table 3 ijms-21-00797-t003:** Summary of SNP markers exhibiting segregation distortion.

Segregation Pattern	Marker Number	To Female Parent	To Male Parent	To Heterozygous	Total	Ratio (%)
lm × ll	13,039	613	224	0	837	6.04
nn × np	12,824	210	573	0	783	5.75
hk × hk	176	0	0	36	36	16.98
Total	26,039	823	797	36	1656	5.98

**Table 4 ijms-21-00797-t004:** Characteristics of the linkage groups of the integrated genetic map.

Linkage Group	Chromosome	Length (cM)	Number of SNPs	Number of Bin Markers	Max Gap (cM)	Gap < 5cM (%)	Mean Markers Distance (cM)
LG1	1	56.80	398	48	5.31	99.75	1.18
LG2	2	83.84	522	82	16.97	99.81	1.02
LG3	3	64.64	477	60	17.51	99.79	1.08
LG4	4	74.97	718	74	14.84	99.86	1.01
LG5	5	96.51	1314	89	9.74	99.77	1.08
LG6	6	46.65	306	55	3.37	100.00	0.85
LG7	7	88.49	1171	119	4.34	100.00	0.74
LG8	8	63.05	465	70	6.77	99.79	0.91
LG9	9	77.43	2698	104	4.34	100.00	0.74
LG10	10	90.05	2509	99	9.74	99.96	0.91
LG11	11	78.52	776	73	15.90	99.49	1.08
LG12	12	80.81	1341	109	4.82	100.00	0.74
LG13	13	92.73	2072	82	13.29	99.86	1.13
LG14	14	92.80	1541	91	14.32	99.87	1.02
LG15	15	91.75	1902	89	13.29	99.90	1.03
LG16	16	89.44	2642	152	2.89	100.00	0.59
LG17	17	14.91	177	20	2.89	100.00	0.75
LG18	18	100.53	3511	126	4.34	100.00	0.80
LG19	19	76.46	1499	120	4.34	100.00	0.64
Total		1460.38	26,039	1662			
Average		76.86	1370.47	87.47	8.90	99.89	0.88

**Table 5 ijms-21-00797-t005:** The results of QTL mapping for berry firmness.

Year	QTL Name	Linkage Group	QTL Peak (cM)	LOD	Explained Variance (%)
2016	*qBF18-2016*	18	68.29	4.98	22.3
2017	*qBF18-2017*	18	60.12	4.88	21.5
2018	*qBF18-2018*	18	60.60	5.86	28.6

**Table 6 ijms-21-00797-t006:** Candidate genes screened from the QTL regions.

Gene ID	Location	Function Description
*VIT_18s0041g00700*	18:25246440-25247384	PREDICTED: NAC domain-containing protein 90-like
*VIT_18s0041g02140*	18:27268849-27284583	PREDICTED: agamous-like MADS-box protein AGL12
*VIT_18s0041g02410*	18:27514537-27525639	PREDICTED: abscisic-aldehyde oxidase-like
*VIT_18s0089g00210*	18:27952657-27956129	PREDICTED: endoglucanase 3

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
