# Peer review of "Construction of a High-Density Genetic Map and Mapping of Firmness in Grapes (*Vitis vinifera* L.) Based on Whole-Genome Resequencing"

_ijms, 2020, doi:10.3390/ijms21030797_

Round 1
Reviewer 1 Report
The accurate revision of the manuscript has been appreciated and most points have been addressed.However, I would like to ask you to revise the discussion related to the point 6 and 7.
Point 6 (inconstancy of QTLs). The new discussion of page 10 and following is biased by the overestimation of the positive effect of the high number of markers. Increasing the number of markers is one of the way to increase the map saturation. However, the high number of markers cannot compensate the low map resolution due to the small pop size. You say to have mapped 26039 markers. I’m sure that very few of these were informative! Please, for each position in the map, keep only one marker and exclude all those that do not give recombination with that marker. This is an information that you should have to report in the paper.
Let me say again that a good map resolution can be achieved either by increasing the number of markers and by increasing the pop size. You have a high number of markers but a too small pop size. In a small pop size recombinations that occur with low frequency between close markers cannot be seen and therefore many markers overlap their genetic position. As a consequence, the QTLs you find have low LOD score and low map resolution and in turn the region you point to with the QTL encompasses many genes, among which you might have had difficulty in finding the candidates.
If your controlled cross yield a small progeny, please stress this fact and the reasons in the results so that the reader can understand why you decided to study the genetic control of berry firmness in so few offsprings.
You state “The effective sequencing depth of the female parent, the male parent, and their progeny was 44.93×, 47.05×, and 11.67–20.22×, respectively, which was sufficient to detect the recombination breakpoints”. Comment: not true. You are not able to see recombination breackpoints at frequencies < 1% with a pop size of 109 individuals. Please write “… which was sufficient to detect an appreciable number of recombination breakpoints”.
Point 7 (candidate genes) The revision is appreciated but, although I’m not expert in the biochemistry of softening of berry flesh, the genes you suggest as potential candidates are weakly supported by experimental evidences. You should have to report the number of genes you considered at each QTL e the reason for which you selected those you report in the manuscript. Hence, you checked the gene expression through qRT-PCR only on the two parents and not in the cross progeny. I would suggest to stop your paper to the QTL analysis without going through the analysis of candidate genes; otherwise, please, describe the relationship between the function of candidate genes and the berry firmness, adding possibly further citations from the literature.
Further suggestions:
Bring each year of table 1 in a single line. Page 13 line 413 “All the primers were designed with Primer Premier 5.0 software…”. Remove the duplicate “Primer”.Author Response
Point 1: The accurate revision of the manuscript has been appreciated and most points have been addressed. However, I would like to ask you to revise the discussion related to the point 6 and 7.
Response 1: Thanks for your suggestion, relevant content has been revised according to the reviewer’s suggestion.
Point 2:Point 6 (inconstancy of QTLs). The new discussion of page 10 and following is biased by the overestimation of the positive effect of the high number of markers. Increasing the number of markers is one of the way to increase the map saturation. However, the high number of markers cannot compensate the low map resolution due to the small pop size. You say to have mapped 26039 markers. I’m sure that very few of these were informative! Please, for each position in the map, keep only one marker and exclude all those that do not give recombination with that marker. This is an information that you should have to report in the paper.
Response 2: Thanks for your suggestion, due to the relatively small population and the large number of SNPs in this study, many SNPs overlap their genetic position, the adjacent SNPs with same genetic distance in an interval were recognized as a single bin marker, the average inter-marker distance was recalculated according to bin markers (line 18-19, page 1; line 147-168, page 5; Figure 3, 7; line 262-267, page 10; Supplementary Materials Table S2 and Table S3).
Point 3:Let me say again that a good map resolution can be achieved either by increasing the number of markers and by increasing the pop size. You have a high number of markers but a too small pop size. In a small pop size recombinations that occur with low frequency between close markers cannot be seen and therefore many markers overlap their genetic position. As a consequence, the QTLs you find have low LOD score and low map resolution and in turn the region you point to with the QTL encompasses many genes, among which you might have had difficulty in finding the candidates.
If your controlled cross yield a small progeny, please stress this fact and the reasons in the results so that the reader can understand why you decided to study the genetic control of berry firmness in so few offsprings.
Response 3: Thanks for your suggestion, we agree with reviewer, the resolution of QTL mapping largely depends on marker density and population size, increasing in population size can improve the QTL precision. It is difficult to generate a hybrid population that is suitable for QTL mapping, such as recombinant inbred lines or F2 population in grape and other fruit trees, which are all highly heterozygous, have long breeding cycles, therefore, the population size are often smaller in fruit research than those of annual crops, the F1 population in fruit study is relatively small, such as Ziziphus jujube (103 progenies ), grape (91 offspring), citrus (110 hybrids), pear (102 individuals). Related information has been added according to the reviewer’s suggestion (line 337-340, page 11).
The references as follow:
Wang, Z., Zhang, Z., Tang, H., Zhang, Q., Zhou, G., Li, X. (2019). High-density Genetic Map Construction and QTL Maping of Phenotype Related Traits in Ziziphus jujuba Mill. Frontiers in Plant Science, 10, 1424. Fu, P., Tian, Q., Lai, G., Li, R., Song, S., Lu, J. (2019). Cgr1, a ripe rot resistance QTL in Vitis amurensis ‘Shuang Hong’grapevine. Horticulture research, 6(1), 1-9. Lima, R. P., Curtolo, M., Merfa, M. V., Cristofani-Yaly, M., & Machado, M. A. (2018). QTLs and eQTLs mapping related to citrandarins’ resistance to citrus gummosis disease. BMC genomics, 19(1), 516. Wu, J., Li, L. T., Li, M., Khan, M. A., Li, X. G., Chen, H., Zhang, S. L. (2014). High-density genetic linkage map construction and identification of fruit-related QTLs in pear using SNP and SSR markers. Journal of experimental botany, 65(20), 5771-5781.Point 4:You state “The effective sequencing depth of the female parent, the male parent, and their progeny was 44.93×, 47.05×, and 11.67–20.22×, respectively, which was sufficient to detect the recombination breakpoints”. Comment: not true. You are not able to see recombination breackpoints at frequencies < 1% with a pop size of 109 individuals. Please write “… which was sufficient to detect an appreciable number of recombination breakpoints”.
Response 4: Many thanks for your suggestions, related information has been revised according to the reviewer’s suggestion (line 262-263, page 10).
Point 5:Point 7 (candidate genes) The revision is appreciated but, although I’m not expert in the biochemistry of softening of berry flesh, the genes you suggest as potential candidates are weakly supported by experimental evidences. You should have to report the number of genes you considered at each QTL e the reason for which you selected those you report in the manuscript. Hence, you checked the gene expression through qRT-PCR only on the two parents and not in the cross progeny. I would suggest to stop your paper to the QTL analysis without going through the analysis of candidate genes; otherwise, please, describe the relationship between the function of candidate genes and the berry firmness, adding possibly further citations from the literature.
Response 5: Thanks for your suggestion, Reviewer 2 recommend supplementing this section, we have made correction and add the citation according to the reviewer’s suggestion (line 330-333, page 11).
Point 6:Bring each year of table 1 in a single line.
Response 6: Thanks for your suggestion, we have brought each year of table in a single line according to the reviewer’s suggestion (line 104, page 3).
Point 7:Page 13 line 413 “All the primers were designed with Primer Premier 5.0 software…”. Remove the duplicate “Primer”.
Response 7: we are sorry for our carelessness, relevant content has been removed according to the reviewer’s suggestion (line 426, page 13).
Special thanks to you for your good comments.
Reviewer 2 Report
The authors improved the manuscript by following the suggestions of the reviewers.
I suggest some further improvements:
In the abstract, indicate the relative annotation near the VIT_ code of the gene.
Fig. 4: please add statistical analysis
In M&M, point 4.8 RNA Isolation and qRT-PCR: use at least two housekeeping genes for the normalization of qRT-PCR data. In addition, please specify the number of biological replicates used and how the “Crimson seedless” data reported in Fig 4 were obtained. That is, if they derive from several plants or from a single one.
Author Response
Point 1:In the abstract, indicate the relative annotation near the VIT_ code of the gene.
Response 1: Thank you for your suggestions, related information has been added in the abstract (line 26-27, page 1)
Point 2:Fig. 4: please add statistical analysis
Response 2: Thank you for your suggestions, relevant statistical analysis has been added according to the reviewer’s suggestion (line 207-209, page 9; line 211-212, page 9).
Point 3:In M&M, point 4.8 RNA Isolation and qRT-PCR: use at least two housekeeping genes for the normalization of qRT-PCR data. In addition, please specify the number of biological replicates used and how the “Crimson seedless” data reported in Fig 4 were obtained. That is, if they derive from several plants or from a single one.?
Response 3: Thank you for your suggestions, The normalization of quantitative real time RT-PCR is important to obtain accurate gene expression data. VvACTIN was relatively stable, so it was often selected as a only reference in grape research (Lin et al. 2019; Khalil-Ur-Rehman, et al. 2017; Vergara et al. 2012). All samples in this section design 3 independent biological replicates, 30 berries from 10 clusters of single plant tree as one biological replicates, RNA extraction after mixing of 30 berries. Relevant statistical analysis has been added according to the reviewer’s suggestion (line 420, 427, page 13).
The references as follow:
Lin, H., Leng, H., Guo, Y., Kondo, S., Zhao, Y., Shi, G., Guo, X. (2019). QTLs and candidate genes for downy mildew resistance conferred by interspecific grape (V. vinifera L.× V. amurensis Rupr.) crossing. Scientia horticulturae, 244, 200-207. Khalil-Ur-Rehman, M., Wang, W., Xu, Y. S., Haider, M. S., Li, C. X., Tao, J. M. (2017). Comparative study on reagents involved in grape bud break and their effects on different metabolites and related gene expression during winter. Frontiers in plant science, 8, 1340. Vergara, R., Parada, F., Rubio, S., Pérez, F. J. (2012). Hypoxia induces H2O2 production and activates antioxidant defence system in grapevine buds through mediation of H2O2 and ethylene. Journal of Experimental Botany, 63(11), 4123-4131.
Special thanks to you for your good comments.